# Influence of Electronic Environment on the Radiative Efficiency of 9-Phenyl-9*H*-carbazole-Based *ortho*-Carboranyl Luminophores

**DOI:** 10.3390/molecules26061763

**Published:** 2021-03-21

**Authors:** Seok Ho Lee, Ji Hye Lee, Min Sik Mun, Sanghee Yi, Eunji Yoo, Hyonseok Hwang, Kang Mun Lee

**Affiliations:** Department of Chemistry, Institute for Molecular Science and Fusion Technology, Kangwon National University, Chuncheon 24341, Korea; chem1st@kangwon.ac.kr (S.H.L.); jhlee81@kangwon.ac.kr (J.H.L.); bbcisgj2000@kangwon.ac.kr (M.S.M.); yshee7410@kangwon.ac.kr (S.Y.); dmswl995655@kangwon.ac.kr (E.Y.); hhwang@kangwon.ac.kr (H.H.)

**Keywords:** *closo*-*ortho*-carborane, 9*H*-carbazole, intramolecular charge transfer, electron-donating group, radiative decay

## Abstract

The photophysical properties of *closo-ortho-*carboranyl-based donor–acceptor dyads are known to be affected by the electronic environment of the carborane cage but the influence of the electronic environment of the donor moiety remains unclear. Herein, four 9-phenyl-9*H*-carbazole-based *closo*-*ortho*-carboranyl compounds (**1F**, **2P**, **3M**, and **4T**), in which an *o*-carborane cage was appended at the C3-position of a 9-phenyl-9*H*-carbazole moiety bearing various functional groups, were synthesized and fully characterized using multinuclear nuclear magnetic resonance spectroscopy and elemental analysis. Furthermore, the solid-state molecular structures of **1F** and **4T** were determined by X-ray diffraction crystallography. For all the compounds, the lowest-energy absorption band exhibited a tail extending to 350 nm, attributable to the spin-allowed π–π* transition of the 9-phenyl-9*H*-carbazole moiety and weak intramolecular charge transfer (ICT) between the *o*-carborane and the carbazole group. These compounds showed intense yellowish emission (λ_em_ = ~540 nm) in rigid states (in tetrahydrofuran (THF) at 77 K and in films), whereas considerably weak emission was observed in THF at 298 K. Theoretical calculations on the first excited states (S_1_) of the compounds suggested that the strong emission bands can be assigned to the ICT transition involving the *o*-carborane. Furthermore, photoluminescence experiments in THF‒water mixtures demonstrated that aggregation-induced emission was responsible for the emission in rigid states. Intriguingly, the quantum yields and radiative decay constants in the film state were gradually enhanced with the increasing electron-donating ability of the substituent on the 9-phenyl group (‒F for **1F** < ‒H for **2P** < ‒CH_3_ for **3M** < ‒C(CH_3_)_3_ for **4T**). These features indicate that the ICT-based radiative decay process in rigid states is affected by the electronic environment of the 9-phenyl-9*H*-carbazole group. Consequently, the efficient ICT-based radiative decay of *o*-carboranyl compounds can be achieved by appending the *o*-carborane cage with electron-rich aromatic systems.

## 1. Introduction

Over the past few decades, π-conjugated aromatic compounds bearing an icosahedral c*loso*-*ortho*-carborane (*closo*-*o*-1,2-C_2_B_10_H_12_) cluster have been extensively investigated [1,2,3,4,5] because of their unique photophysical properties and reasonable electrochemical stabilities [1,3,6,7,8]. Owing to these features, such *o*-carboranyl compounds have recently been proposed as promising optoelectronic materials for organic light emitting diodes [7,8] and organic thin-film transistors [9,10]. The intriguing photophysical properties of *o*-carboranyl luminophores originate from the electronic donor‒acceptor (D–A) dyad formed by combining a π-conjugated aromatic organic fluorophore (donor) with an *o*-carborane (acceptor) [4,7,8,9,10,11,12,13,14,15,16,17,18,19,20,21,22,23,24,25,26,27,28,29,30,31,32,33,34,35,36]. The strong electron-withdrawing ability of the carbon atoms in the *o*-carborane cage, which is derived from the high polarizability of the σ-aromaticity [37,38,39,40,41], allows this moiety to act as an electron acceptor during external excitation and relaxation processes. As a result, intramolecular charge transfer (ICT) can be induced between aromatic groups and the *o*-carborane cage [2,4,11,12,13,14,15,16,17,18,19,20,21,22,23,24,25,26,27,28,29,30,31,32,33,34,35,36]. The luminescence behavior of *o*-carborane-based D‒A dyads is characterized by the radiative decay of the ICT transition [2,4,7,8,10,11,12,13,14,15,16,17,18,19,20,21,22,23,24,25,26,27,28,29,30,31,32,33,34,35,36].

Recent research has shown that the ICT-based luminescence characteristics of D‒A dyads are dramatically affected by the electronic environment of the *o*-carborane cage. For example, the photophysical properties are drastically altered by the deboronation of a *closo*-*o*-carborane to give a *nido*-*o*-carborane (nest-like structure, where one boron atom is removed from the icosahedron) [33,42,43,44,45,46,47,48,49], as the anionic character of the *nido*-type structure interrupts the ICT transition [42,43,44,45,46,47,48,49,50,51,52,53,54]. The intrinsic differences between the electronic properties of *closo*- and *nido*-*o*-carboranes have been investigated, providing inspiration as novel molecular scaffolds for chemodosimeter materials. However, the effect of the electronic environment of the π-conjugated aryl group itself (donor moiety) in D–A dyads on the photophysical properties has only rarely been investigated.

Thus, to gain in-depth insight into how the electronic properties of the π-aryl group in *o*-carboranyl D–A dyads influence the photophysical properties, especially the ICT-based emission characteristics, we strategically designed and prepared four *o*-carboranyl compounds based on 9-phenyl-9*H*-carbazole (Figure 1), which is a widely used *N*-heterocyclic donor moiety. The *o*-carborane cage was appended at the C3-position of the 9*H*-carbazole moiety and various functional groups (‒F, ‒H, ‒CH_3_, and ‒C(CH_3_)_3_) were introduced at the *para*-position of the 9-phenyl ring to modify the electronic effects. Subsequently, the photophysical properties were examined and theoretical calculations were performed on the ground (S_0_) and excited (S_1_) states of these compounds to determine the impact of the electronic environment of the 9*H*-carbazole group on the ICT-based radiative decay efficiencies of *o*-carboranyl D–A dyads.

## 2. Materials and Methods

### 2.1. General Considerations

All experiments were carried out under an inert N_2_ atmosphere using standard Schlenk and glove box techniques. Anhydrous solvents (toluene and trimethylamine (NEt_3_); Sigma-Aldrich) were dried by passing each solvent through an activated alumina column. Spectrophotometric-grade solvents (tetrahydrofuran (THF), dichloromethane (DCM), methanol (MeOH), and *n*-hexane; Alfa Aesar (Haverhill, MA, USA)) were used as received. Commercial reagents were used as received from Sigma-Aldrich (St. Louis, MO, USA) (phenylacetylene, 3-bromocarbazole, 3-bromo-9-phenyl-9*H*-carbazole, 3-bromo-9-(*p*-tolyl)-9*H*-carbazole, 1-fluoro-4-iodobenzene, 1-bromo-4-*tert*-butylbenzene, bis(triphenylphosphine)palladium(II) dichloride (Pd(PPh_3_)_2_Cl_2_), copper(I) iodide (CuI), tripotassium phosphate (K_3_PO_4_), *trans*-1,2-diaminocyclohexane, diethyl sulfide (Et_2_S), magnesium sulfate (MgSO_4_), and poly(methyl methacrylate) (PMMA)) and Alfa Aesar (decaborane (B_10_H_14_)). The deuterated solvent (dichloromethane-*d*_2_ (CD_2_Cl_2_); Cambridge Isotope Laboratories (Tewksbury, MA, USA)) was dried over activated molecular sieves (5 Å). Nuclear magnetic resonance (NMR) spectra were recorded on a Bruker Avance 400 spectrometer (400.13 MHz for ^1^H and ^1^H{^11^B}, 100.62 MHz for ^13^C, 376.50 MHz for ^19^F, and 128.38 MHz for ^11^B{^1^H}) (Bruker Corporation, Billerica, MA, USA) at ambient temperature. Chemical shifts are given in ppm and are referenced against external tetramethylsilane (Me_4_Si) (^1^H, ^1^H{^11^B}, and ^13^C), trichlorofluoromethane (CCl_3_F) (^19^F), and BF_3_·Et_2_O (^11^B{^1^H}). Elemental analysis was performed on an EA3000 analyzer (Eurovector, Pavia, Italy) at the Central Laboratory of Kangwon National University.

### 2.2. General Synthetic Procedure for Acetylene Precursors (**CzA**, **2PA**, and **3MA**)

The acetylene precursors (**CzA**, **2PA**, and **3MA**) were synthesized as follows using the appropriate amounts of starting materials. Toluene and NEt_3_ (2/1, *v*/*v*) were added via cannula to a mixture of bromocarbazole compound, CuI, and Pd(PPh_3_)_2_Cl_2_ at 25 °C. After stirring the mixture for 30 min, phenylacetylene (2.0 equiv. with respect to the bromocarbazole starting material) was added to the resulting dark brown slurry. The reaction mixture was then refluxed at 120 °C for 24 h. The volatiles were removed by rotary evaporation to afford a dark brown residue. The solid residue was purified by column chromatography on silica gel (eluent: DCM/*n*-hexane = 1/9, *v*/*v*) to produce a solid acetylene precursor.

#### 2.2.1. Data for CzA

3-Bromocarbazole (1.2 g, 5.0 mmol), CuI (95 mg, 0.50 mmol), Pd(PPh_3_)_2_Cl_2_ (0.35 g, 0.50 mmol), and phenylacetylene (1.1 mL, 10 mmol) afforded **CzA** as a white solid. Yield = 30% (0.40 g). ^1^H NMR (CD_2_Cl_2_): δ 8.29 (s, 1H, ‒N*H*), 8.27 (t, *J* = 2.1, 1H), 8.07 (d, *J* = 7.8 Hz, 1H), 7.59 (dd, *J* = 8.2, 4.0 Hz, 1H), 7.56 (dd, *J* = 7.8, 4.0 Hz, 2H), 7.46 (m, 1H), 7.43 (td, *J* =7.4, 2.0 Hz, 2H), 7.36 (td, J = 12.2, 4.8 Hz, 3H), 7.25 (td, *J* = 8.0, 1.5 Hz, 1H). ^13^C NMR (CD_2_Cl_2_): δ 139.93, 139.27, 131.39, 129.39, 128.45, 127.94, 126.45, 123.87, 123.82, 123.35, 122.74, 120.42, 119.95, 113.88, 110.90, 110.87, 90.59 (acetylene-*C*), 87.42 (acetylene-*C*). Anal. Calcd for C_20_H_13_N: C, 89.86; H, 4.90; N, 5.24. Found: C, 89.29; H, 4.58; N, 5.01.

#### 2.2.2. Data for 2PA

3-Bromo-9-phenyl-9*H*-carbazole (1.6 g, 5.0 mmol), CuI (95 mg, 0.50 mmol), Pd(PPh_3_)_2_Cl_2_ (0.35 g, 0.50 mmol), and phenylacetylene (1.1 mL, 10 mmol) afforded **2PA** as a yellow solid. Yield = 41% (0.70 g). ^1^H NMR (CD_2_Cl_2_): δ 8.35 (s, 1H), 8.15 (d, *J* = 7.8 Hz, 1H), 7.63 (t, *J* = 8.4 Hz, 2H), 7.59 (d, *J* = 3.6 Hz, 1H), 7.57 (t, *J* = 2.4 Hz, 3H), 7.56 (m, 1H), 7.50 (tt, *J* = 8.2, 2.4 Hz, 1H), 7.43 (dd, *J* = 8.0, 2.0 Hz, 1H), 7.42 (m, 1H), 7.39 (d, *J* = 3.4 Hz, 2H), 7.35 (m, 2H), 7.32 (td, *J* = 4.9, 1.4 Hz, 1H). ^13^C NMR (CD_2_Cl_2_): δ 141.41, 140.57, 137.21, 131.42, 130.04, 129.49, 128.47, 127.98, 127.84, 127.05, 126.55, 123.85, 123.79, 123.43, 122.83, 120.47, 120.45, 114.35, 110.04, 109.98, 90.53 (acetylene-*C*), 87.73 (acetylene-*C*). Anal. Calcd for C_26_H_17_N: C, 90.93; H, 4.99; N, 4.08. Found: C, 90.49; H, 4.67; N, 3.95.

#### 2.2.3. Data for 3MA

3-Bromo-9-(*p*-tolyl)-9*H*-carbazole (1.2 g, 3.4 mmol), CuI (65 mg, 0.34 mmol), Pd(PPh_3_)_2_Cl_2_ (0.24 g, 0.34 mmol), and phenylacetylene (0.75 mL, 6.8 mmol) afforded **3MA** as a yellow solid. Yield = 27% (0.33 g). ^1^H NMR (CD_2_Cl_2_): δ 8.36 (s, 1H), 8.16 (d, *J* = 7.7 Hz, 1H), 7.60 (d, *J* = 7.8 Hz, 3H), 7.43 (s, 4H), 7.41 (d, *J* = 1.8 Hz, 1H), 7.39 (s, 2H), 7.37 (d, *J* = 4.8 Hz, 2H), 7.34 (t, 4.2 Hz, 1H), 7.31 (d, *J* = 7.6 Hz, 1H). 2.49 (s, 3H, ‒C*H*_3_). ^13^C NMR (CD_2_Cl_2_): δ 141.59, 140.74, 137.98, 134.49, 131.45, 130.62, 129.48, 128.50, 127.99, 126.86, 126.52, 123.87, 123.35, 122.77, 120.44, 120.35, 114.22, 110.09, 110.01, 90.69 (acetylene-*C*), 87.74 (acetylene-*C*), 21.06 (‒*C*H_3_). Anal. Calcd for C_27_H_19_N: C, 90.72; H, 5.36; N, 3.92. Found: C, 90.29; H, 5.10; N, 3.77.

### 2.3. Synthesis of **1FA**

Toluene (10.0 mL) was added via cannula to a mixture of **CzA** (0.30 g, 1.1 mmol), CuI (23 mg, 0.12 mmol), K_3_PO_4_ (0.48 g, 2.2 mmol), and *trans*-1,2-diaminocyclohexane (14 mg, 0.12 mmol) at 25 °C. After stirring for 10 min, 1-fluoro-4-iodobenzene (0.13 mL, 1.1 mmol) was added. Then, the reaction mixture was refluxed at 120 °C for 24 h. After cooling to 25 °C, the solvent was removed under vacuum to afford a dark brown crude product. Purification through column chromatography on silica gel (eluent: DCM/*n*-hexane = 1/4, *v*/*v*) gave **1FA** as a white solid. Yield = 35% (0.14 g). ^1^H NMR (CD_2_Cl_2_): δ 8.36 (s, 1H), 8.16 (d, *J* = 7.8 Hz, 1H), 7.61 (q, *J* = 2.2 Hz, 1H), 7.60 (dd, *J* = 2.4, 0.8 Hz, 2H), 7.52 (q, *J* = 8.2 Hz, 2H), 7.44 (d, 7.8 Hz, 1H), 7.39 (d, *J* = 4.2 Hz, 2H), 7.36 (m, 2H), 7.32 (t, *J* = 4.8 Hz, 3H), 7.30 (d, *J* = 2.4 Hz, 1H). ^13^C NMR (CD_2_Cl_2_): δ 163.11, 160.65, 141.57, 140.72, 133.24, 133.21, 131.47, 129.62, 129.10, 129.01, 128.52, 128.05, 126.67, 123.93, 123.82, 123.42, 122.82, 120.60, 120.54, 117.08, 116.85, 114.54, 109.86, 109.80, 90.56 (acetylene-*C*), 87.88 (acetylene-*C*). ^19^F NMR (CD_2_Cl_2_): δ −113.78. AnalAnal. Calcd for C_26_H_16_FN: C, 86.41; H, 4.46; F, 5.26; N, 3.88. Found: C, 86.20; H, 4.25; N, 3.79.

### 2.4. Synthesis of **4TA**

A procedure analogous to that for **1FA** was employed utilizing **CzA** (0.27 g, 1.0 mmol), CuI (19 mg, 0.10 mmol), K_3_PO_4_ (0.42 g, 2.0 mmol), *trans*-1,2-diaminocyclohexane (11 mg, 0.10 mmol), and 1-bromo-4-*tert*-butylbenzene (0.17 mL, 1.2 mmol). Purification through column chromatography on silica gel (eluent: DCM/*n*-hexane = 1/7, v/v) gave **4TA** as a white solid. Yield = 52% (0.21 g). ^1^H NMR (CD_2_Cl_2_): δ 8.35 (s, 1H), 8.16 (d, *J* = 7.7 Hz, 1H), 7.65 (d, *J* = 8.5 Hz, 2H), 7.58 (d, *J* = 8.0 Hz, 3H), 7.49 (d, *J* = 8.3 Hz, 2H), 7.43 (t, *J* = 8.0 Hz, 3H), 7.39 (t, *J* = 8.0 Hz, 3H) 7.31 (t, *J* = 8.2 Hz, 1H), 1.43 (s, 9H, ‒C(C*H*_3_)_3_). ^13^C NMR (CD_2_Cl_2_): δ 151.32, 141.83, 141.00, 134.73, 131.73, 129.75, 128.80, 128.29, 127.27, 126.80, 124.14, 123.64, 123.05, 120.74, 120.65, 114.47, 110.45, 110.39, 90.90 (acetylene-*C*), 87.98 (acetylene-*C*), 35.09 (‒*C*(CH_3_)_3_), 31.51 (‒C(*C*H_3_)_3_). Anal. Calcd for C_30_H_25_N: C, 90.19; H, 6.31; N, 3.51. Found: C, 89.73; H, 6.09; N, 3.42.

### 2.5. General Synthetic Procedure for Carbazole-Based o-Carborane Compounds (**1F**, **2P**, **3M**, and **4T**)

To a toluene solution (50 mL) of B_10_H_14_ (1.2 equiv. with respect to the acetylene precursor) and acetylene precursor (**1FA**, **2PA**, **3MA**, and **4TA**) was added an excess amount of Et_2_S (3.0 equiv. with respect to the acetylene precursor) at ambient temperature. After heating to reflux, the reaction mixture was stirred for a further 72 h. The solvent was removed under vacuum and MeOH (50 mL) was added. The resulting yellow solid was filtered and redissolved in toluene. The solution was purified by column chromatography on silica gel (eluent: DCM/*n*-hexane = 1/4, *v*/*v*) to produce a white solid carborane compound.

#### 2.5.1. Data for 1F

**1FA** (0.14 g, 0.39 mmol), B_10_H_14_ (57 mg, 0.47 mmol), and Et_2_S (0.13 mL, 1.2 mmol) afforded **1F** as a white solid. Yield = 27% (51 mg). ^1^H{^11^B} NMR (CD_2_Cl_2_): δ 8.25 (s, 1H), 8.08 (d, *J* = 8.0 Hz, 1H), 7.52 (td, *J* = 8.8, 1.2 Hz, 3H), 7.42 (q, *J* = 6.4 Hz, 3H), 7.27 (t, *J* = 8.0 Hz, 4H), 7.16 (d, *J* = 7.8 Hz, 1H), 7.11 (t, *J* = 7.9 Hz, 2H), 7.07 (d, *J* = 8.8 Hz, 1H), 3.46 (br s, 2H, CB-B*H*), 2.62 (br s, 3H, CB-B*H*), 2.56 (br s, 3H, CB-B*H*), 2.34 (br s, 2H, CB-B*H*). ^13^C NMR (CD_2_Cl_2_): δ 163.16, 160.70, 141.77, 141.56, 132.84, 132.81, 130.91, 130.83, 130.20, 128.97, 128.89, 128.56, 128.34, 126.96, 123.34, 123.00, 122.69, 122.46, 120.76, 120.43, 117.09, 116.86, 109.97, 109.11, 87.45 (CB-*C*), 86.19 (CB-*C*). ^11^B{^1^H} NMR (CD_2_Cl_2_): δ −3.48 (br s, 3B), −9.95 (br s, 3B), −11.65 (br s, 4B).^19^F NMR (CD_2_Cl_2_): δ −113.52. Anal. Calcd for C_26_H_26_B_10_FN: C, 65.11; H, 5.46; N, 2.92. Found: C, 64.89; H, 5.25; N, 2.81.

#### 2.5.2. Data for 2P

**2PA** (0.70 g, 2.0 mmol), B_10_H_14_ (0.30 g, 2.5 mmol), and Et_2_S (0.66 mL, 6.1 mmol) afforded **2P** as a white solid. Yield = 34% (0.32 g). ^1^H{^11^B} NMR (CD_2_Cl_2_): δ 8.25 (d, *J* = 1.9 Hz, 1H), 8.08 (d, *J* = 7.8 Hz, 1H), 7.57 (t, *J* = 8.2 Hz, 2H), 7.52 (t, *J* = 8.0 Hz, 2H), 7.43 (m, 5H), 7.33 (d, *J* = 7.8 Hz, 1H), 7.28 (t, *J* = 6.8 Hz, 1H), 7.16 (d, *J* = 7.8 Hz, 1H), 7.11 (t, *J* = 7.9 Hz, 3H), 3.46 (br s, 2H, CB-B*H*), 2.63 (br s, 2H, CB-B*H*), 2.55 (br s, 4H, CB-B*H*), 2.34 (br s, 2H, CB-B*H*). ^13^C NMR (CD_2_Cl_2_): δ 141.60, 130.89, 130.81, 130.18, 130.02, 128.44, 128.31, 127.96, 126.87, 126.84, 123.29, 123.01, 122.72, 122.30, 120.64, 120.36, 110.16, 109.26, 87.51 (CB-*C*), 86.17 (CB-*C*). ^11^B{^1^H} NMR (CD_2_Cl_2_): δ −3.58 (br s, 3B), −10.02 (br s, 3B), −11.75 (br s, 4B). Anal. Calcd for C_26_H_27_B_10_N: C, 67.65; H, 5.90; N, 3.03. Found: C, 67.33; H, 5.80; N, 2.89.

#### 2.5.3. Data for 3M

**3MA** (0.32 g, 0.90 mmol), B_10_H_14_ (0.13 g, 1.1 mmol), and Et_2_S (0.29 mL, 2.7 mmol) afforded **3M** as a white solid. Yield = 44% (0.19 g). ^1^H{^11^B} NMR (CD_2_Cl_2_): δ 8.24 (s, 1H), 8.07 (d, *J* = 7.8 Hz, 1H), 7.52 (d, *J* = 8.0 Hz, 2H), 7.48 (d, *J* = 7.2 Hz, 1H), 7.38 (m, 3H), 7.28 (t, *J* = 6.8 Hz, 4H), 7.16 (d, *J* = 7.4 Hz, 1H), 7.10 (t, *J* = 7.9 Hz, 3H), 3.45 (br s, 2H, CB-B*H*), 2.61 (br s, 2H, CB-B*H*), 2.54 (br s, 4H, CB-B*H*), 2.44 (s, 3H, ‒C*H*_3_), 2.33 (br s, 2H, CB-B*H*). ^13^C NMR (CD_2_Cl_2_): δ 141.75, 138.15, 130.90, 130.81, 130.57, 130.17, 128.38, 128.29, 126.76, 126.66, 123.25, 122.88, 122.62, 122.12, 120.47, 120.31, 110.16, 109.25, 87.59 (CB-*C*), 86.17 (CB-*C*), 20.99 (‒*C*H_3_). ^11^B{^1^H} NMR (CD_2_Cl_2_): δ −3.52 (br s, 3B), −9.91 (br s, 3B), −11.72 (br s, 4B). Anal. Calcd for C_27_H_29_B_10_N: C, 68.18; H, 6.15; N, 2.94. Found: C, 68.04; H, 6.05; N, 2.88.

#### 2.5.4. Data for 4T

**4TA** (0.21 g, 0.52 mmol), B_10_H_14_ (76 mg, 0.62 mmol), and Et_2_S (0.17 mL, 1.6 mmol) afforded **4T** as a white solid. Yield = 31% (80 mg). ^1^H{^11^B} NMR (CD_2_Cl_2_): δ 8.25 (s, 1H), 8.09 (d, *J* = 7.7 Hz, 1H), 7.59 (d, *J* = 8.2 Hz, 2H), 7.53 (d, *J* = 7.8 Hz, 2H), 7.49 (d, *J* = 7.8 Hz, 1H), 7.41 (t, *J* = 7.4 Hz, 1H), 7.34 (t, *J* = 8.2 Hz, 3H), 7.29 (t, *J* = 7.4 Hz, 1H), 7.19 (t, *J* = 7.8 Hz, 1H) 7.12 (m, 3H), 3.46 (br s, 1H, CB-B*H*), 2.62 (br s, 3H, CB-B*H*), 2.55 (br s, 4H, CB-B*H*), 2.33 (br s, 2H, CB-B*H*), 1.40 (s, 9H, ‒C(C*H*_3_)_3_). ^13^C NMR (CD_2_Cl_2_): δ 151.47, 142.02, 141.81, 134.35, 131.21, 131.14, 130.50, 128.68, 128.63, 127.26, 127.08, 126.59, 123.59, 123.21, 122.96, 122.44, 120.81, 120.64, 110.56, 109.64, 87.90(CB-*C*), 86.48(CB-*C*), 35.06 (‒*C*(CH_3_)_3_), 31.45 (‒C(*C*H_3_)_3_). ^11^B{^1^H} NMR (CD_2_Cl_2_): δ −3.74 (br s, 3B), −9.94 (br s, 3B), −11.80 (br s, 4B). Anal. Calcd for C_30_H_35_B_10_N: C, 69.60; H, 6.81; N, 2.71. Found: C, 69.48; H, 6.72; N, 2.61.

### 2.6. UV/Vis Absorption and Photoluminescence (PL) Measurements

Solution-phase UV/Vis absorption and PL measurements for each *o*-carborane compound were performed in degassed THF using a 1 cm quartz cuvette (30 µM) at 298 K. PL measurements were also carried out in THF at 77 K, in THF/water mixtures, and in the film state (5 wt% doped in PMMA on a 15 × 15 mm quartz plate (thickness = 1 mm)). The UV−vis absorption and PL spectra were recorded on Jasco V-530 (Jasco, Easton, MD, USA) and FluoroMax-4P spectrophotometers (HORIBA, Edison, NJ, USA), respectively. The absolute PL quantum yields (Φ_em_) for the THF/water mixture and film samples were obtained using an absolute PL quantum yield spectrophotometer (FM-SPHERE, 3.2-inch internal integrating sphere on FluoroMax-4P, HORIBA) at 298 K. Fluorescence decay lifetimes of the films were measured at 298 K using a time-correlated single-photon counting (TCSPC) spectrometer (FLS920, Edinburgh Instruments, Livingston, UK) at the Central Laboratory of Kangwon National University. The TCSPC spectrometer was equipped with a pulsed semiconductor diode laser excitation source (EPL, 375 ps) and a microchannel plate photomultiplier tube (MCP-PMT, 200–850 nm) detector.

### 2.7. X-ray Crystallography

Each single X-ray quality crystal of **1F** and **4T** was grown from a DCM/*n*-hexane mixture. The single crystals were coated with Paratone oil and mounted on a glass capillary. Crystallographic measurements were performed using a Bruker D8QUEST diffractometer with graphite monochromated Mo-Kα radiation (λ = 0.71073 Å) and a CCD area detector. The structures of **1F** and **4T** were determined by direct methods, and all non-hydrogen atoms were subjected to anisotropic refinement with a full-matrix least-squares method on *F*^2^ using the SHELXTL/PC software package. The X-ray crystallographic data for **1F** and **4T** are available in CIF format (CCDC 2,065,228 for **1F** and 2,065,228 for **4T**), provided free of charge by The Cambridge Crystallographic Data Centre. Hydrogen atoms were placed at their geometrically calculated positions and refined using a riding model on the corresponding carbon atoms with isotropic thermal parameters. The detailed crystallographic data are given in Appendix A.

### 2.8. Computational Calculations

The optimized geometries for the ground (S_0_) and first excited (S_1_) states of all the *o*-carboranyl compounds in THF were obtained at the B3LYP/6-31G(d,p) [55] level of theory. The vertical excitation energies at the optimized S_0_ geometries as well as the optimized geometries of the S_1_ states were calculated using time-dependent density functional theory (TD-DFT) [56] at the same level of theory. Solvent effects were evaluated using the self-consistent reaction field (SCRF) based on the integral equation formalism of the polarizable continuum model (IEFPCM) with THF as the solvent [57]. All geometry optimizations were performed using the Gaussian 16 program [58]. The percent contribution of a group in a molecule to each molecular orbital was calculated using the GaussSum 3.0 program [59].

## 3. Results and Discussion

### 3.1. Synthesis and Characterization

9-Phenyl-9*H*-carbazole-based *o*-carboranyl compounds (**1F**, **2P**, **3M**, and **4T**), in which the *o*-carborane cage is appended at the C3-position of the carbazole moiety, were synthesized as shown in Figure 1. Sonogashira coupling reactions between phenylacetylene and bromocarbazole precursors (3-bromocarbazole, 3-bromo-9-phenyl-9*H*-carbazole, and 3-bromo-9-(*p*-tolyl)-9*H*-carbazole) produced phenylacetylene-substituted carbazole compounds (**CzA**, **2PA**, and **3MA**, respectively) in moderated yields (27–41%). Further, Ullmann coupling reactions of **CzA** with 1-fluoro-4-iodobenzene and 1-bromo-4-*tert*-butylbenzene produced acetylene precursors **1FA** and **4TA**, respectively, in yields of 35% and 52%. The 9-phenyl-9*H*-carbazole-based *o*-carboranyl compounds were prepared via cage-forming reactions with B_10_H_14_ using **1FA**, **2PA**, **3MA**, or **4TA** in the presence of Et_2_S (yields of 27–44%) [60,61,62]. All of the precursors and prepared *o*-carboranyl carbazole compounds (**1F**, **2P**, **3M**, and **4T**) were fully characterized using multinuclear (^1^H, ^1^H{^11^B}, ^13^C, and ^11^B{^1^H}, ^19^F) NMR spectroscopy (Appendix A in the Appendix A) and elemental analysis. In particular, the ^1^H{^11^B} NMR spectra of the *o*-carboranyl carbazole compounds exhibited resonances corresponding to the 9-phenyl-9*H*-carbazole moiety and terminal phenyl groups in the region of 8.3–7.0 ppm. Further, broad singlet peaks at 3.5–2.3 ppm (corresponding to 10 H atoms) confirmed the existence of –BH units in the *closo*-*o*-carborane cages. Two sharp signals were observed at approximately 88 and 86 ppm in the ^13^C NMR spectra, which were attributed to the two carbon atoms of the *closo*-*o*-carboranyl groups. In addition, three broad singlet peaks were observed between −3 and −12 ppm in the ^11^B{^1^H} NMR spectra of all the *closo*-*o*-carborane compounds, which clearly confirmed the presence of the *o*-carborane cage. The molecular structures of **1F** and **4T** was also determined by X-ray crystallography (Figure 2; detailed parameters, including selected bond length and angles, are provided in Appendix A). The crystal structures of both **1F** and **4T** revealed the carbazole moiety to be perfectly planar, as evidenced by the sum of the three C–N–C angles (∑[C–N–C] = 359.1° for **1F**, and 359.5° for **4T**, Appendix A), which indicates that each N atom center is *sp*^2^ hybridized and that all the atoms in the carbazole moiety show aromaticity.

### 3.2. Experimental and Theoretical Analysis of Photophysical Properties

The photophysical properties of the 9-phenyl-9*H*-carbazole-based *o*-carboranyl compounds (**1F**, **2P**, **3M**, and **4T**) were investigated using UV/Vis-absorption and PL spectroscopies (Figure 3 and Table 1). All the compounds exhibited a low absorption band centered at λ_abs_ = ~329 nm with a broad shoulder extending to 350 nm. This absorption band was mainly attributed to the spin-allowed π–π* local excitation (LE) transition of the 9-phenyl-9*H*-carbazole moiety, as the parent compound (9-phenyl-9*H*-carbazole) exhibited a similar major absorption band in the region of λ_abs_ = 327–337 nm (Appendix A). However, TD-DFT calculations for the S_0_ state of the *o*-carboranyl compounds suggested that this band could also be attributed to a weak ICT transition from the carbazole moiety to the *o*-carborane cage and terminal phenyl ring (vide infra). Furthermore, a strong absorption peak centered at λ_abs_ = 279 nm was observed in the spectrum of each *o*-carboranyl compound, which originates from the π–π* transition of the carbazole group, as 9-phenyl-9*H*-carbazole showed an absorption maximum at λ_abs_ = 284 nm (Appendix A).

To obtain insight into the origin of the electronic transitions for **1F**, **2P**, **3M**, and **4T**, TD-DFT calculations were performed on each S_0_-optimized structure [56]. These calculations were based on the solid-state molecular structure of **1F** and an IEFPCM was used to include the effect of THF as the solvent [57]. The computational results for the S_0_-optimized structures revealed that the major low-energy electronic transitions were mainly associated with transitions from the highest occupied molecular orbital (HOMO) to the lowest unoccupied molecular orbital (LUMO) (Figure 4). The HOMO of each compound was predominantly localized on the carbazole (>97% in each compound, Appendix A), whereas the LUMO was distributed over the *o*-carborane (~34% in each compound) as well the carbazole (>41%) and terminal phenyl group (>24%). These calculation results indicate that the lowest-energy electronic transition for the carbazole-based *o*-carboranyl compounds originates from both the π–π* LE transition of the appended carbazole moiety and an ICT transition from the carbazole moiety to the *o*-carborane cage and terminal phenyl ring.

The emission properties of **1F**, **2P**, **3M**, and **4T** were investigated under various conditions using PL measurements (Figure 3 and Table 1). Remarkably, all the *o*-carboranyl compounds exhibited very weak emission in the region of 380–420 nm in THF at 298 K, whereas intense emission at λ_em_ = ~535 nm was observed in THF at 77 K. In comparison, the emission of 9-phenyl-9*H*-carbazole was centered at λ_em_ = 361 and 377 nm (Appendix A), which verifies that the faint emission in THF at 298 K can be attributed to an LE transition of the 9-phenyl-9*H*-carbazole moiety. Furthermore, the emission of the *o*-carboranyl compounds at 77 K was significantly red-shifted compared with that of 9-phenyl-9*H*-carbazole, indicating that this transition corresponded to ICT involving the *o*-carborane (vide infra). Such differences in the emission features at 298 and 77 K typically result from structural changes being restricted in the rigid state; for example, inhibiting the elongation of the C‒C bond in the *o*-carborane cage is known to prevent the ICT-based radiative decay mechanism [12,13,14,16,23,24,33,34,35,63,64,65]. Indeed, the calculated lengths of the C‒C bonds for the *o*-carborane cages in the S_1_ state were much longer (2.39 Å for all compounds) than those in the S_0_ state (~1.8 Å) as well as the experimentally measured values for **1F** (1.73 Å, Appendix A) and **4T** (1.74 Å) based on the X-ray crystal structures.

The PL spectra of the four *o*-carboranyl compounds in the film state (5 wt% doped in PMMA) also displayed distinct emission centered at λ_em_ = ~545 nm (Table 1), resulting in intense yellowish emission (inset, Figure 3), as this rigid solid state restricted the elongation of C‒C bonds. Moreover, the origin of the yellow emission in the film state was investigated by measuring the PL of the four *o*-carboranyl compounds in a THF–water mixture (30 μM) (Figure 5 and Table 1). The low-energy emission centered at ~560 nm was drastically enhanced as the water fraction increased (*f*_w_). Consequently, the most aggregated state in THF/water (*f*_w_ = 90%) exhibited intense yellowish emission (λ_em_ = 552–559 nm; inset, Figure 5 and Table 1), similar to that observed in the film state. These observations are characteristic of aggregation-induced emission (AIE) phenomena. Consequently, the remarkably enhanced emission of the carbazole-based *o*-carboranyl compounds in the film state could originate from a strong AIE effect as well as an increase in the efficiency of the ICT-based radiative decay owing to structural rigidity. In addition, the absolute quantum yields (Φ_em_) of the four *o*-carboranyl compounds in the film state were more than two times greater than those in THF/water (*f*_w_ = 90%) mixtures (34% and 10% for **1F**, 44% and 20% for **2P**, 51% and 35% for **3M**, and 61% and 47% for **4T**, respectively, Table 1), which supported the existence of ICT-based emission as well as AIE in the film state.

The calculation results for the S_1_-optimized structures of the *o*-carboranyl compounds indicated that the major transition associated with the low-energy emission involves a HOMO → LUMO transition (Figure 4). The LUMO of each compound is mostly localized on the *o*-carborane cage (>74%, Appendix A), whereas the HOMO is predominantly localized on the 9-phenyl-9*H*-carbazole group (>93%). These results strongly suggest that the emission observed in the rigid states (THF at 77 K and film) mainly originates from a radiative decay process based on ICT between the *o*-carborane and carbazole moieties. Consequently, the electronic transitions of each *o*-carboranyl compound were precisely predicted using computational methods.

### 3.3. Electronic Effect on ICT-Based Radiative Decay Efficiency

The Φ_em_ and decay lifetime (τ_obs_) of each *o*-carboranyl compound (**1F**, **2P**, **3M**, and **4T**) in the film state were investigated to gain insight into the influence of electronic effects on the radiative decay efficiency of the ICT transition. Intriguingly, the Φ_em_ values of the *o*-carboranyl compounds in the film were gradually enhanced (34% for **1F**, 44% for **2P**, 51% for **3M**, and 55% for **4T**; Table 1) as the electron-donating effect of the substituent on the 9-phenyl ring increased (electron-donating ability: ‒F of **1F** < ‒H of **2P** < ‒CH_3_ of **3M** < ‒C(CH_3_)_3_ of **4T**). The τ_obs_ values for all the *o*-carboranyl compounds were similar (5.2–6.8 ns) (Table 1 and Appendix A), indicating fluorescent characteristics. A comparison of the radiative (*k*_r_; Table 1) and nonradiative (*k*_nr_) decay constants of the *o*-carboranyl compounds, as calculated using Φ_em_ and τ_obs_, demonstrated a distinct difference in the efficiency of the ICT-based radiative process for each *o*-carboranyl compound in the film state. The *k*_r_ values of all the *o*-carboranyl compounds in the film state were enhanced (from 5.0 × 10^7^ s^−1^ for **1F** to 1.5 × 10^8^ s^−1^ for **4T**) in accordance with the increasing electron-donating effect of the substituent, whereas all the *k*_nr_ values were similar (9.3–9.7 × 10^7^ s^−1^). This distinct trend verifies that the efficiency of the radiative decay process corresponding to the ICT transition involving the *o*-carborane unit is strongly affected by the electronic characteristics of the substituents on the 9-phenyl group of the 9*H*-carbazole moiety. These findings suggest that an electron-rich carbazole moiety, obtained by introducing an electron-donating group, can accelerate the ICT-based radiative decay pathway.

## 4. Conclusions

Herein, we evaluated the impact of electron-donating effects on the ICT-based radiative process in 9-phenyl-9*H*-carbazole-based *o*-carboranyl compounds (**1F**, **2P**, **3M**, and **4T**) bearing various functional groups on the 9-phenyl group of the carbazole moiety. Although the *o*-carboranyl compounds were weakly emissive in solution at 298 K, an intense emission corresponding to an ICT transition involving the *o*-carborane unit were observed in rigid states (in solution at 77 K and in the film state). PL measurements in THF‒water mixtures suggested that an AIE effect was also involved in the emission in the film state. Intriguingly, a gradual increase in the Φ_em_ and *k_r_* values for the ICT-based emission of the *o*-carboranyl compounds in the film state was observed as the electron-donating ability of the substituent on the 9-phenyl group increased (‒F < ‒H < ‒CH_3_ < ‒C(CH_3_)_3_). These properties strongly indicate that the ICT-based radiative decay process in the *o*-carboranyl compounds was induced by the electron-rich nature of the appended aromatic group. Consequently, the results of this study suggest that the electronic environment of an aryl group linked to an *o*-carborane unit can control the efficiency of radiative decay processes based on ICT transitions.

## Figures and Tables

**Figure 1 molecules-26-01763-f001:**
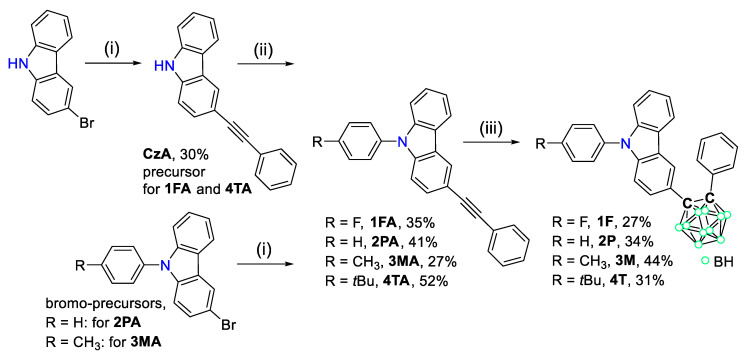
Synthetic routes for 9-phenyl-9*H*-carbazole-based *o*-carboranyl compounds (**1F**, **2P**, **3M**, and **4T**). Reaction conditions: (i) phenylacetylene, CuI, Pd(PPh_3_)_2_Cl_2_, toluene/NEt_3_ (2/1, *v*/*v*), 120 °C, 24 h; (ii) CuI, K_3_PO_4_, *trans*-1,2-diaminocyclohexane, 1-fluoro-4-iodobenzene (for **1FA**) or 1-bromo-4-*tert*-butylbenzene (for **4TA**), 120 °C, 24 h; (iii) B_10_H_14_, Et_2_S, toluene, 120 °C, 72 h.

**Figure 2 molecules-26-01763-f002:**
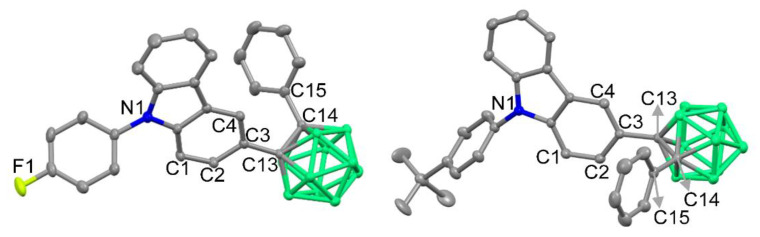
X-ray crystal structures for **1F** (left) and **4T** (right) (50% thermal ellipsoids with H atoms omitted for clarity).

**Figure 3 molecules-26-01763-f003:**
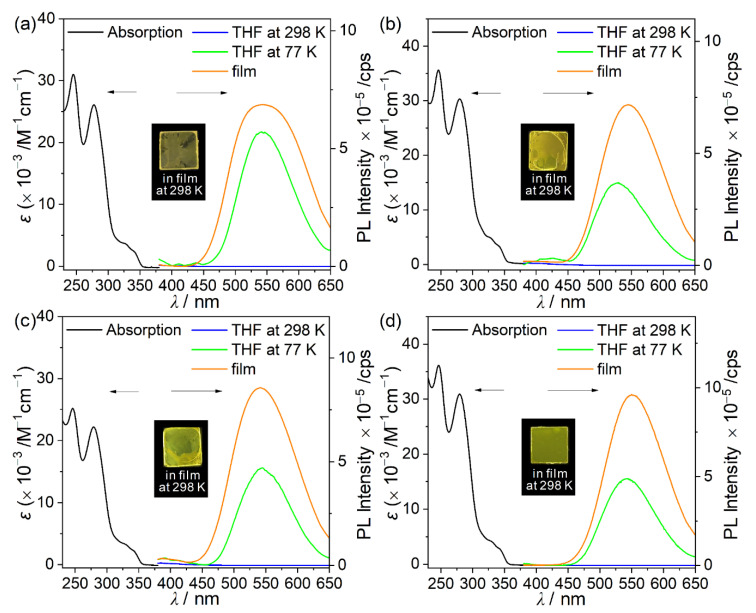
UV-vis absorption (left) and PL spectra (right) of (**a**) **1F** (λ_ex_ = 329 nm), (**b**) **2P** (λ_ex_ = 328 nm), (**c**) **3M** (λ_ex_ = 330 nm), and (**d**) **4T** (λ_ex_ = 333 nm). Black line: absorption spectra in THF (30 μM), blue line: PL spectra in THF (30 μM) at 298 K, green line: PL spectra in THF (30 μM) at 77 K, and orange line: PL spectra in film (5 wt% doped in PMMA) at 298 K. Insets: emission color in the film state under irradiation by a hand-held UV lamp (λ_ex_ = 365 nm).

**Figure 4 molecules-26-01763-f004:**
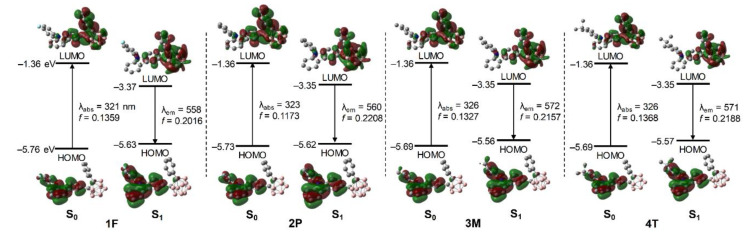
Frontier molecular orbitals for **1F**, **2P**, **3M**, and **4T** in the ground state (S_0_) and first excited singlet state (S_1_) with relative energies from DFT calculations (isovalue 0.04). The transition energy (in nm) was calculated using the TD-B3LYP method with 6-31G(d) basis sets.

**Figure 5 molecules-26-01763-f005:**
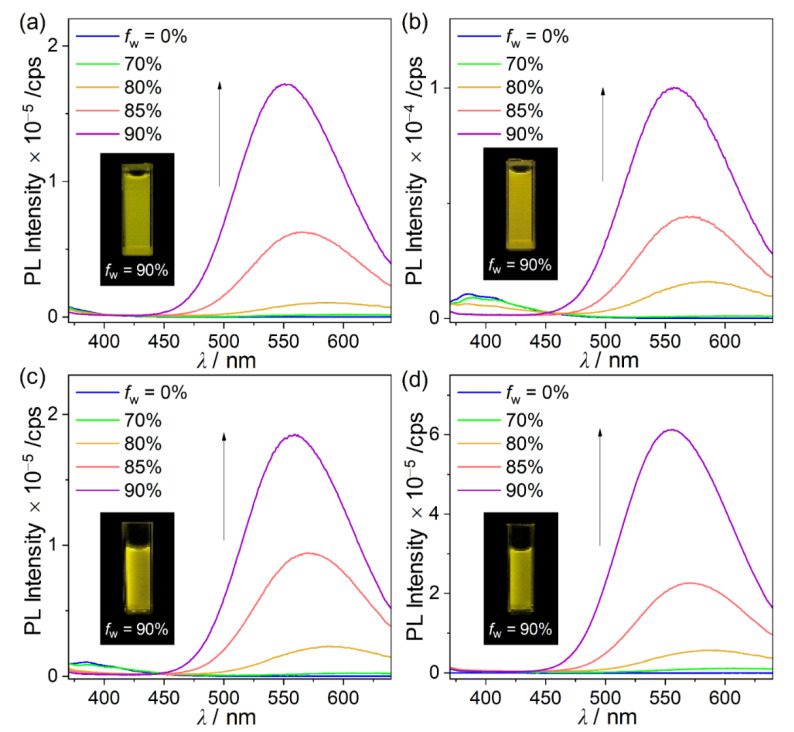
PL spectra of (**a**) **1F** (λ_ex_ = 329 nm), (**b**) **2P** (λ_ex_ = 328 nm), (**c**) **3M** (λ_ex_ = 330 nm), and (**d**) **4T** (λ_ex_ = 333 nm) in THF/water mixtures (30 µM). Insets: emission color at *f*_w_ = 90% under irradiation by a hand-held UV lamp (λ_ex_ = 365 nm).

**Table 1 molecules-26-01763-t001:** Photophysical data for 9-phenyl-9*H*-carbazole-based *o*-carboranyl compounds.

Compd.	*λ*_abs_^1^/nm (ε × 10^−3^ M^−1^ cm^−1^)	*λ*_ex_/nm	*λ*_em_/nm			
			THF^2^	77 K^1^	film^3^	*f*_w_ = 90%^4^
**1F**	329 (3.5), 279 (26.1)	329	-^8^	541	543	552
**2P**	328 (4.9), 279 (30.3)	328	-^8^	528	545	557
**3M**	330 (3.2), 279 (22.2)	330	-^8^	544	542	559
**4T**	333 (3.8), 279 (30.9)	333	-^8^	543	549	556
Compd.	Φ_em_^5^		τ_obs_^3^/ns	*k*_r_^3,6^/× 10^8^ s^−1^		*k*_nr_^3,7^/× 10^7^ s^−1^
	film^3^	*f*_w_ = 90%^4^				
**1F**	0.34	0.10	6.8	0.50		9.7
**2P**	0.44	0.20	5.9	0.75		9.5
**3M**	0.51	0.35	5.2	0.98		9.4
**4T**	0.61	0.47	4.2	1.5		9.3

^1^30 μM in THF. ^2^30 μM, observed at 298 K. ^3^Measured in the film state (5 wt% doped in PMMA). ^4^30 μM in a THF/water mixture (1/9, *v*/*v*), observed at 298 K. ^5^Absolute PL quantum yield. ^6^*k*_r_ = Φ_em_/τ_obs_. ^7^*k*_nr_ = *k*_r_(1/Φ_em_ − 1). ^8^Not observed due to weak emission.

## Data Availability

Data is contained within the article or Appendix A.

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
