# Peer review of "Influence of Electronic Environment on the Radiative Efficiency of 9-Phenyl-9*H*-carbazole-Based *ortho*-Carboranyl Luminophores"

_molecules, 2021, doi:10.3390/molecules26061763_

Round 1

Reviewer 1 Report

In this manuscript, Kang Mun Lee and coworkers investigated the influence of electronics on the radiative efficiency of 9-phenyl-9H-carbazole-based ortho-carbonyl luminophores. Through studies were performed by varying the electron-donating ability on the 9-phenyl group. All prepared compounds were well characterized by 1H and 13C NMR spectroscopies, and X-ray crystallography. The authors found that the quantum yields and radioactive decay constants in the film state were gradually enhanced with the increasing electron donating nature of the substituents. Considering the importance of efficient ICT-based radiative decay of ortho-carboranyl compounds and the performed studies on them, the publication in Molecules is recommended.

Author Response

In this manuscript, Kang Mun Lee and coworkers investigated the influence of electronics on the radiative efficiency of 9-phenyl-9H-carbazole-based ortho-carbonyl luminophores. Through studies were performed by varying the electron-donating ability on the 9-phenyl group. All prepared compounds were well characterized by 1H and 13C NMR spectroscopies, and X-ray crystallography. The authors found that the quantum yields and radioactive decay constants in the film state were gradually enhanced with the increasing electron donating nature of the substituents. Considering the importance of efficient ICT-based radiative decay of ortho-carboranyl compounds and the performed studies on them, the publication in Molecules is recommended.

Response: We thank the reviewer for such a favorable evaluation.

Reviewer 2 Report

This is a very nice paper describing the synthesis and characterization of novel carborane-based fluorescent dyads based on carbazole/carborane framework. The focus of examining the electronics of the carbazole is clearly defined and well executed. Th synthetic work is competently done and the spectroscopic and computational studies clearly presented. Given the emphasis on the tuning of the electronic properties by changing the R substituent, it might have been better served by using a more diverse range of functional groups which better span both electron-withdrawing and electron-releasing groups, e.g. alkoxy or dimethylamino as strong pi-donor groups while cyano, nitro or acetyl are strongly activating (based on Hammett sigma-p parameters).  Notwithstanding this comment, the work is competently done and, with the exception of a few minor changes, is suitable for publication in Molecules.

Minor Comments:

Abstract, line 14 "...540 nm) in rigid states..." (please insert units)

Section 2.1: end of last paragraph. "...are referenced against external tetramethylsilane..." (not tetramethyl tin(IV)!!)

Section 2.7: "the structures of 1F and 4T were determined by direct methods..." (they were not assessed). Also please update the SI: The data collection method is not multi-scan; multi-scan is the absorption correction method. The data collection method stated in the cif file is "phi and omega scans"

Section 3.2:

(a) The authors comment that the system exhibits a low energy absorption band around 329 nm with a broad shoulder and vibronic features out to 350 nm. I am not convinced that these are vibronic in nature; the SI indicates the lowest energy transition (HOMO-LUMO) at 321 nm with a series of low-lying electronic excited states ranging down to 279 nm for 1F. This sentence should be corrected.

(b) "...which likely originates..." I think the authors can be more assertive here based on their calculations and indicate "...which originates..."

Author Response

This is a very nice paper describing the synthesis and characterization of novel carborane-based fluorescent dyads based on carbazole/carborane framework. The focus of examining the electronics of the carbazole is clearly defined and well executed. Th synthetic work is competently done and the spectroscopic and computational studies clearly presented. Given the emphasis on the tuning of the electronic properties by changing the R substituent, it might have been better served by using a more diverse range of functional groups which better span both electron-withdrawing and electron-releasing groups, e.g. alkoxy or dimethylamino as strong pi-donor groups while cyano, nitro or acetyl are strongly activating (based on Hammett sigma-p parameters).  Notwithstanding this comment, the work is competently done and, with the exception of a few minor changes, is suitable for publication in Molecules.

Response: We thank the reviewer for such a favorable evaluation.

Minor Comments:

Abstract, line 14 "...540 nm) in rigid states..." (please insert units)

Response: As the reviewer pointed out, we revised the part (highlighted in yellow).

Section 2.1: end of last paragraph. "...are referenced against external tetramethylsilane..." (not tetramethyl tin(IV)!!)

Response: As the reviewer pointed out, we revised the part (highlighted in yellow).

Section 2.7: "the structures of 1F and 4T were determined by direct methods..." (they were not assessed). Also please update the SI: The data collection method is not multi-scan; multi-scan is the absorption correction method. The data collection method stated in the cif file is "phi and omega scans"

Response: As the reviewer pointed out, we revised the part in the manuscript (highlighted in yellow) and Table S1 of the Supplementary Material.

Section 3.2:

(a) The authors comment that the system exhibits a low energy absorption band around 329 nm with a broad shoulder and vibronic features out to 350 nm. I am not convinced that these are vibronic in nature; the SI indicates the lowest energy transition (HOMO-LUMO) at 321 nm with a series of low-lying electronic excited states ranging down to 279 nm for 1F. This sentence should be corrected.

Response: As the reviewer pointed out, we revised the part to "a broad shoulder extending to 350 nm" (highlighted in yellow).

(b) "...which likely originates..." I think the authors can be more assertive here based on their calculations and indicate "...which originates..."

Response: As the reviewer pointed out, we revised the part (highlighted in yellow).

Reviewer 3 Report

Reviewer comments included in the attached file

Author Response

The paper deals with the functionalization of carboranes with carbazole derivatives in order to produce more photo physically efficient D-A dyads. My area of expertise does not allow me to review the synthetic procedures and as so I will concentrate on the X-ray, Photophysics and quantum calculations. The paper is difficult to read and the discussion is hard to follow and should be improved preferentially with tables and schemes of the ongoing processes. In my opinion the paper is suitable for publication after some major points are addressed:

1 – B3LYP is not an adequate functional for Photophysics, since it lacks the appropriate asymptotical behavior of charge density. From my experience non range separated functionals produce absorption and emission simulated spectra blue shifted (typically 1 eV) when compared to experimental spectra. Even CAMB3LYP (the LR equivalent of B3LYP is still typically 8% of to the blue side). The authors should explain their choice of functional.

Response: Thank you for this valuable question. We agree with your point that the CAMB3LYP functional could be more suitable than the B3LYP functional to analyzing the photophysical properties (electronic transitions) of organic and organometallic luminophores. However, many researchers, who investigate the photophysical properties of organic/organometallic luminophores, particularly o-carboranyl compounds, have already used the B3LYP functional to confirm the absorption and emission behaviors (e.g., Ref. 7, 9, 12, 14, 15, 18, 23‒29, 31, 33‒36, 39, 40, 44‒46, 48). Furthermore, these calculation results were relatively well-matched with the experimental ones, particularly the emissive features. Therefore, we believe that the calculation results presented in this manuscript can sufficiently verify the experimental result for o-carboranyl compounds.

2 – Figure S16 reports a well-defined absorption and PL spectra of a simple carbazole. I would like the author to provide a benchmark of B3LYP by including the equivalent simulated spectra.

Response: As per your suggestion, the calculation results for 9-phenyl-9H-carbazole have been inserted in Figure S16.

3 – The Stokes shift observed is to big to have PL originating in an S1 relaxed species electronically identical to the S1 in absorption. As the authors mention this PL is probably originating in AIM. This is INCOMPATIBLE with their calculations. The relaxed S1 structures have huge oscillator strengths (single molecule TDDFT) and, if that is true, why the PL is absent in THF?

Response: As we had described in Section 3.2. Experimental and Theoretical Analysis of Photophysical Properties (page 7, bottom part), the PL is absent in THF at 298 K because the elongation of the C‒C bonds in the o-carborane cage prevents ICT-based radiative decay [corresponding literatures: Ref 12‒14,16,23,24,33‒35,63‒65], resulting in the quenching of the PL in THF at 298 K. In addition, the origin of emissive characteristics of o-carboranyl compounds is fundamentally radiative decay through ICT-based transition between o-carborane (acceptor) and appended aromatic ring (donor), although all the compounds exhibit AIE behavior. The fact that these ICT-based emissions became enhanced in rigid state (in THF at 77 K and film state) also verify that the AIE phenomenon arises from the reinforcement of ICT-based emission in rigid (solid) state. All the features, including the large Stokes shift, are generally observed in many o-carboranyl luminophores, as has been reported in previous decades. Therefore, we regretfully do not agree with the reviewer’s comment.

4 – The emissive state should be electronically different from the LE excited state and only observed in aggregates. The SI should include date for the excited states that are (for instance) within 1 eV of S1.

Response: As we mentioned above, the fact that these ICT-based emissions were enhanced in rigid state (in THF at 77 K and film state) verify that the AIE phenomenon arises from the reinforcement of ICT-based emission in rigid (solid) state. It is further proven by the fact that the emission regions are significantly similar to each other in THF at 77 K, in the film state, and for fw = 80%. Consequently, the major transitions of the four o-carboranyl compounds are HOMO → LUMO transitions, i.e., ICT transition (as indicated by the calculation results for the o-carboranyl compounds), although these compounds showed significantly faint LE-based emission.

5 – The authors have made poor utilization of the x-ray structures. The figure bellow shows myanalysis of the packing of one of the structures: The picture depicts a dimeric structure in the solid state which can easily be retained in films or bad solvents. In my opinion this is the emissive structure with a large red shift. The interaction between the S1 states of each monomer generate two new states one dark and one highly stabilized bright state. Calculations should be carried out in this dimmer to prove me right or wrong.

Response: As we mentioned above, the AIE phenomenon arises from the reinforcement of ICT-based emission in rigid (solid) state because the emission regions in THF at 77 K, in film state, and for fw = 80% are significantly similar to each other. Consequently, the emissive characteristics for the o-carboranyl compounds in rigid state (film and THF at 77 K) are not revealed by the intermolecular interactions between the single molecules, but simply by the inhibition of the intramolecular structure variation (elongation of the C‒C bond in the o-carborane cage and rotation of the cages). Therefore, we regretfully do not agree with the reviewer’s comment that the calculation results for the dimeric system of o-carboranyl compounds are needed.

Round 2

Reviewer 3 Report

Although not totally convinced by the authors reply I do agree that they provided sound scientific justifications for their choices. As so I don't see any objection for publication of this work in its present form.